# Comparison between the Test and Simulation Results for PLA Structures 3D Printed, Bending Stressed

**DOI:** 10.3390/molecules26113325

**Published:** 2021-06-01

**Authors:** Dorin Catana, Mihai-Alin Pop, Denisa-Iulia Brus

**Affiliations:** 1Department of Materials Engineering and Welding, Transilvania University of Brasov, 500036 Brasov, Romania; 2Department of Materials Science, Transilvania University of Brasov, 500036 Brasov, Romania; mihai.pop@unitbv.ro; 3School for Doctoral Studies in Socio-Humanities, Dunarea de Jos University, 800008 Galati, Romania; denisa04.c@gmail.com

**Keywords:** additive manufacturing, poly(lactic acid), finite element analysis (FEA), simulation, optimization

## Abstract

The additive manufacturing process is one of the technical domains that has had a sustained development in recent decades. The designers’ attention to equipment and materials for 3D printing has been focused on this type of process. The paper presents a comparison between the results of the bending tests and those of the simulation of the same type of stress applied on 3D-printed PLA and PLA–glass structures. The comparison of the results shows that they are close, and the simulation process can be applied with confidence for the streamline of filament consumption, with direct consequences on the volume and weight of additive manufactured structures. The paper determines whether the theories and concepts valid in the strength of materials can be applied to the additive manufacturing pieces. Thus, the study shows that the geometry of the cross-section, by its shape (circular or elliptical) and type (solid or ring shaped), influences the strength properties of 3D-printed structures. The use of simulation will allow a significant shortening of the design time of the new structures. Moreover, the simulation process was applied with good results on 3D-printed structures in which two types of filaments were used for a single piece (structure).

## 1. Introduction

Currently, the technological processes of material manufacturing are numerous and diversified. Some of them are old and have been used by humans since the Bronze or Iron Age, and others are more recent. Whether we are talking about the new or old technologies, researchers have permanent and sustained concerns for improving them, increasing energy efficiency, and reducing environmental pollution. The recent studies show that the largest weight of the processing technologies belongs to subtractive (classical) technologies. Since the early 1980s, a new technology was implemented, known as additives manufacturing or 3D printing, that is based on the principle of adding material. This technology has developed rapidly in a relatively short period of time, concomitant with increasing the number of materials that can be processed by using it. Moreover, the number of applications based on additive manufacturing has continuously increased, but often these did not meet the expectations of potential users. Being a new technology, it was found that the influence of the processing parameters on the properties of 3D-printed pieces (parts) have many aspects that need to be studied to reach to the degree of reproducibility of the classical technologies. The researchers’ attention is focused on finding possible correlations between the parameters of the printing process and the behaviour to different mechanical stresses of additive manufactured structures. This correlation is necessary to ensure optimized geometries for the manufactured structures, but also for the possibility of predictability of the mechanical and technological properties for 3D-printed pieces. The material on which the studies presented in the paper were made is poly (lactic acid) (PLA), a thermoplastic aliphatic polyester made from renewable resources. The material being biodegradable in the presence of oxygen makes it attractive (interesting) for medical applications. The attractiveness of processing PLA products is also supported by the fact that each stage of the life cycle has recycling technology.

Researchers are trying to find solutions to problems that appear during the printing and use of 3D-printed poly (lactic acid) structures. These concerns are present in numerous scientific articles.

The static test can give similar results with those obtained in a real bending test. It is possible to obtain the optimization of the 3D-printed PLA element based on a bending test and the use of simulation [1]. The researchers studied the influence of processing parameters on the PLA properties, 3D-printed by fused-deposition modelling (FDM). The conclusions of the study show that the layer thickness and infill density are very important for strength value. Moreover, ductility is mainly affected by infill and heat treatment [2]. The finite element method was used to study the behaviour of a 3D-printed hollow spherical part obtained of poly (lactic acid). The researches show that the finite element analysis (FEA) method generates results comparable with the results experimentally obtained. Moreover, the analysis of results showed that the main influence on the elasticity of the 3D-printed part was the printing temperature. On the other hand, the increase in the wall thickness determines an increase in the spherical part elasticity [3]. The researchers conclude that the 3D printing process involves a large number of conflicting parameters. For this reason, it is hard to predict the mechanical properties [4]. Moreover, the behavior of the composite based on PLA and hydroxyapatite (HA) in a simulated body fluids (SBF) chamber was studied. In this environment, the composite had a ductile and intensive to cracks behavior. This evolution is not in line with the brittle behavior observed in conventional tests [5]. In the medical field, many studies were done for the graft optimization. Using a finite element method, the mechanical properties of PLA can be improved for mandibular reconstruction [6]. The studies reveal that it is possible to apply the FEA method for the prediction of the behavior of the 3D-printed PLA, when the structure is uniaxial loaded. Deviation between the experimental and simulation results’ error was under 6.7% [7].

Another study shows that poly (lactic acid) can replace acrylonitrile butadiene styrene (ABS), because PLA is eco-friendly due to having less emissions and wastages than those caused by the classical manufacturing process [8]. In an article, it is shown that PLA has a double asymmetry in its tensile and compressive behaviour. One of the asymmetries is in strength, and the other asymmetry is the constructive behaviour. These show that, in the case of PLA, it is possible to apply a bimodular elasticity model. Moreover, the paper showed that the simulation of the compression is sensitive to the boundary conditions applied at the ends of the specimen (friction, etc.) [9]. The researches show large differences between the simulation results and the practical test results, in the case of many 3D-printed materials. A major influence on the results’ inconsistency belongs to infill grid characteristics (value) [10]. For the PLA parts, when the layer height increases, it appears strength is diminished by 11%. The conclusion was that finer layers lead to better mechanical properties and surface roughness. Moreover, increasing the infill up to 50% greatly improves mechanical strength by 27% [11]. The experimental results show the orthotropic behaviour of FDM specimens printed from the ABS and PLA materials. Moreover, a reduction of the mechanical characteristics can be noted when the raster angle is increased [12]. The tensile strength and failure strain of 3D-printed specimens from the analysed material were shown to significantly vary as a function of build orientation. The tensile strength of these 3D FDM parts can vary from 46 to 85% of reported values for injection-moulded parts [13]. A comprehensive study shows the influence of the process parameters (infill speed, infill density, and infill pattern) on mechanical and physical properties of 3D-printed structures. The parts printed with 100% infill density provide the highest Young’s modulus. When the infill density decreases, the strength of the printed parts decreases. In the infill speed range 70–110 mm/s, the value of 90 mm/s for infill speed gave the optimum tensile. Moreover, a value of 215 °C was found for the PLA filament printing temperature [14]. When PLA is reinforced with short carbon fibres (PLA + CF) (of a length of about 60 µm in a weight fraction of 15%), the tensile modulus (respective to the printing direction) increased about 2.2 times. In addition, the tensile modulus (transverse to the printing direction) and the shear modulus (respective to the plane of printing) increased, respectively, 1.25 and 1.16 times [15].

The PLA-based samples were made by 3D printing with porosity of 30 vol.% and average pore size of 700 µm. Bending of the printed layers led to the destruction of the scaffold material. The trials show that the value was stabilized with a load of 21 MPa in the case of PLA/HA scaffolds, and 18 MPa in the case of PLA. The conclusion is that a porous scaffold can function at such stress for a long time without change [16]. By creating an adequate space frame lattice and shell, it is possible to create the finite element model to predict the linearly elastic response. In this way, for the 3D-printed structures, some mechanical properties can be predicted with reasonable accuracy. Moreover, the finite element analysis method is not computationally expensive [17]. Based on the mechanical testing, finite element modelling (FEM), and macro deformation behavior, the stress–strain relationship was established. This approach was applied with success on high-density polyethylene (HDPE) specimens and it permits the use of the FEM method to mimic closely the mechanical behaviour of polymers during the necking process [18].

The aim of the paper is to verify the possibility of applying finite element analysis for 3D-printed parts from PLA and PLA–Glass, and whether the results obtained by simulation are in line with the results established by tests. The analysis of the concordance between the tests’ results and those of the simulation performed on mechanically stressed structures 3D-printed from PLA and PLA–Glass is the preliminary step to their optimization. By validating the simulation results, the conditions for the optimization of 3D structures are created and represent a necessary and important step in increasing the efficiency of the additive manufacturing. The paper presents the bending behaviour of 3D-printed structures (parts) made of poly (lactic acid) for different geometries of the cross (transversal) section. Moreover, based on the previously established mechanical properties and the results of bending tests, digital modelling and simulation were used to study the bending behaviour and optimize 3D-printed structures.

## 2. Experimental Setup

The first part of the paper showed how researchers found the solution to implement technical concepts known for improving the properties of additive manufactured parts. The materials used in the study are poly (lactic acid) and poly (lactic acid) mixed with glass fibres, more precisely, PLA filament (product code PLA-S3 DPLA 01-285-1000) and PLA–Glass (product code PLA–Glass 041-285-750). For both filaments, technical characteristics of the filaments are given in technical data sheets. These filaments were used in the 3D printing of the specimens (samples) used in the bending tests.

The previous studies on the same materials have shown there are printing parameters which lead to the highest values for the main mechanical properties. The printing of the specimens was performed on a Creat Bot DX-3D double-nozzle printer, and the parameters of the printing process were:Layer height—0.2 mm;Printing temperature—210 °C;Print speed—50 mm/s;Printing angle (overhang angle for support)—45 °C;Bed temperature—61 °C;Infill—100% (the internal structure is solid); also, solid infill at top and bottom;Infill overlap—10%;Infill flow—110%.

The number of the perimeter’s shell was 2 and it had 1 mm thickness (in total) on the horizontal, and the top and bottom each had a thickness of 0.8 mm (4 layers). 

The type of printed specimens were bars or tubes (solid or ring-shaped), having a circular or elliptical geometry of the cross-section. The cross-sections had different sizes. The following filaments were used in the 3D printing of the specimens:PLA (100% PLA filament);PLA-Glass (100% PLA–Glass filament);PLA and PLA–Glass (PLA arranged on the specimen outside in proportion of 31% and PLA–Glass on the inside in proportion of 69%);PLA–Glass and PLA (PLA–Glass arranged on the specimen outside in proportion of 31% and PLA on the inside in proportion of 69%).

The additive manufactured specimens from the presented filaments were subsequently tested at bending. In the printed specimens, in addition to the material, the dimensions of the cross-sections and their shape have been changed. To facilitate the notations and an easy understanding of the types of additive manufactured specimens, Table 1 presents the code assigned to them.

For bar-type specimens with a circular section, the size represents the diameter, and, for the elliptical section, it represents the length of the axes. In the case of tube-type (ring-shaped) specimens with a circular section, the dimensions represent the exterior (outer) and interior (inner) diameter, and, for the specimens with an elliptical section, the dimensions represent the length of the exterior and interior axis (large axis). More precisely, for the elliptical-tube-type specimens, the dimensions of the outside were 18 × 8 mm (the length of the big and of the small axis), and, for the inside, the dimensions were 16 × 6.25 mm.

The number of 3D-printed specimens was 5 for each specimen code. The specimens manufactured by 3D printing were tested for bending. The specimens had a total length of 220 mm, and the distance between the supports was 180 mm for all the performed tests. The loading mode of the test piece is shown in Figure 1 and, due to the low speed of application of the force, the type test is static loading. Figure 1 shows how the bending test was performed, because this must be replicated by simulation. The specimen rests on two cylindrical supports, and the force is applied halfway between them. The contact between the specimen and support is made at the intersection of two straight lines that is punctiform (point-like). Basically, the transmission of the loads was done through points.

The bending tests were performed on a WDW-150S universal testing machine (the applied test force can be modified between 0.1 and 150 kN), and the test conditions were as follows:bending (loading) speed 10 mm/min;stress speed 10 MPa/s.

For each tested specimen at bending, the records provided by the test equipment allowed the load–extension curve (see Figure 2) and stress–strain curve to be established.

Figure 2 represents the stress–strain curve that is returned by the testing machine after the bending tests were performed. From these graphs, primary characteristics of 3D-printed specimens, or values for certain mechanical properties, are collected that will be necessary in the simulation process.

For the simulation process, it is necessary to establish the key material properties required by the simulation program used. These properties were:
Density of the 3D-printed specimen, which is not equal to the density of the filament; it depends on 3D structure’s printed parameters;Modulus of elasticity that was provided by the bending tests (GPa);Yield strength (MPa);Bending deflection (mm);Ultimate strength at bending (bending strength) (MPa);Poisson’s ratio.

Some of these properties have been obtained in tests performed during previous research [19].

## 3. Results and Discussions

Figure 2 shows the rate of curve for a fragile material that does not accumulate high energy or significant deformations until breaking. This was verified by measuring the temperature of the specimen in the force application area. Figure 3 shows the evolution of the temperature (temperature field) close to the end of the test. Because, in the specimen, the temperature increase is only present in the area of force application, it means that, before breaking, the specimen does not accumulate much energy, so the rupture of the specimen is fragile. As can be seen, the temperature in the mentioned area is 22.4 °C for the PLA filament and 23.5 °C for the PLA–Glass filament.

Knowing that the initial temperature of the specimens was 20 °C, the increase in temperature did not exceed a maximum of 4 °C. The area of the specimen which recorded the temperature rise is small, and is located near the place of application of the force. Moreover, the areas adjacent to those mentioned did not register temperature increases. In other words, PLA and PLA–Glass 3D-printed parts behave similarly to alloys, which means that certain concepts valid for alloys can be used with good results for the 3D-printed parts.

After performing the bending tests, the aspect of the breaking area was also studied. For the printed specimens of the PLA or PLA–Glass filament, their fracture (rupture) took place in the area where the force was applied (in the middle of the specimen), and the appearance of the rupture is almost flat (plan). The section appears as if the respective area had been cut, and the surface has a rough (harsh) appearance. Significant differences occur at the roughness of the broken surface which, for PLA, is lower compared to that of PLA–Glass. The behavior is similar for circular or elliptical specimens. In the case of tube-type test specimens, there are some differences. For PLA specimens, the bending more generates the sample’s crash, not its fracture (breaking). Moreover, the initiation of the rupture takes place in the middle of the test tube, and the roughness of the fracture surface for PLA–Glass specimens is higher compared to PLA specimens.

The bending tests performed allowed the value of the strength at bending (bending strength) to be determined for the tested specimens in different variants of filament, geometry, and size of the cross-section and type of specimen (bar or tube). Simultaneously with the determination of the bending strength, the tests also provided the values of the force and the bending deflection (displacement on the Y axis) at which the rupture of the specimen occurred. Combining this information with that obtained in the previous research [19] performed on the same types of filaments and parameters as the current printed specimens, the conditions for using the finite element analysis method are fulfilled to perform the simulation of the bending test. To apply the simulation process, it is necessary to use a model, but its design is important because reproducing the accuracy of the real loading situation will influence the precision of the obtained results. Therefore, in the design stage of the model, attention must be focused on the precise description of the geometry, its constraints, the properties of the material, and the applied loads on it. Using a computer aided design software (CAD), the models on which the bending simulation would be applied were designed. The model has the same geometry and dimensions as the reproduced specimen. The precise (exact) and correct modelling of the real situation allows the simulation process to reproduce, as well as possible, the situation during the bending tests performed on the testing equipment.

The place where the force is applied is located in the middle of the specimen, this being the area that records the highest value for bending deflection. Due to force that acts on the specimen, its deformation takes place, and, at a certain moment, it breaks. The displacement recorded by the specimen during the test is called the bending deflection (displacement), and is a parameter that is measured during the bending test. The value recorded by the test machine for movement is from the upper surface of the specimen.

After performing the bending tests, the obtained results were processed and analysed. Based on the collected results in the tests, it was possible to simulate the bending of the 3D-printed structures. For each type of 3D-printed specimen, performing the simulation meant, in the first stage, the allocation of the proper physical–mechanical properties for the 3D-printed material structure (density, modulus of elasticity, yield strength, Poisson’s ratio). Moreover, the value of the force applied on the specimen during the simulation process was, in fact, the maximum force which produced the rupture of that type of specimen during the bending test. In other words, for both the bending test and the model for which the simulation was applied, the conditions were identical. Because the tests performed during the bending test and the bending simulation were performed under the same conditions, the results obtained should be at least very close, if not identical. The results recorded during the tests were the ultimate strength and the maximum bending deflection, measured in the middle of the distance between the supports. The simulation was performed under the following conditions: study—linear static, mesh type—tetrahedral, and mesh size between 1.21 and 2.75 mm (depending on analysis structure).

By applying the simulation on the designed model, the value of the Von Mises stress and the bending deflection (total deformation) in the point of application of the force were determined. If the bending modelling has been designed correctly, the results should be similar with those obtained by the bending tests. Figure 4 shows the result of the bending simulation for the P_20 specimen. The figure analysis shows that maximum stresses occur in the force application area. The simulation results for this specimen also allowed the determination of the maximum value for the Von Mises stress.

Figure 5 shows the area in the specimen where the stresses that are equal with the ultimate strength value recorded for the specimen during the bending test is located. It can be seen that this area is only found in the force application area. This type of figure allows a better exploration of the stresses developed in the specimen, which facilitates the detection of critical areas in 3D structures. Thus, the geometric shape of the parts can be corrected locally, so that they support the applied stresses.

Figure 6 shows the evolution of the displacements recorded during the simulation in the specimen for which the finite element analysis method was applied. The maximum value for displacement was in the force application area. Comparing the values of the analyzed characteristics determined by simulation with those recorded during the bending test, it is found that they are close.

The simulation process was applied similarly for all types of 3D-printed specimens. Table 2 shows the results of the bending tests for the 3D-printed specimens, but, also, the results obtained by applying the simulation for the respective specimens. In the same way as the one presented, the simulation was applied for the tube-type specimen. The results obtained for this type of test specimen are also presented in Table 2.

In addition, for the tube specimens, the values of the simulation process are close to those recorded during the bending tests.

To easily understand the results from Table 2, they are also presented in a graphical form, because, in this way, the analysis of the behavior of the 3D-printed structures bending tested is facilitated.

Figure 7 presents a comparison between bending strength values obtained by test and simulation. For the bending strength, the differences between test values and simulation values are ±7.24. However, for 10 results the differences are ±5.

Regarding the total deformation, Table 2 (Deformation column) shows high differences between the test and simulation values for the tube-type specimens. For these specimens, the thickness of the specimen wall is less than 2 mm, and they have a very high flexibility. In the table, six values were registered with high differences. For these specimens, some adjustments must be considered in the modelling of the simulation process. 

By additive manufacturing, 3D specimens were printed through the combination in each specimen of the two filament types: PLA and PLA–Glass. The specimens are of the bar-type, with a circular or elliptical cross-section. The 3D-printed specimens have PLA filament on the outside and, in the central part of the section, the PLA–Glass filament was used, but, also, specimens in which the arrangement of the two filaments is reversed (vice versa) are used. Since, in the specimen’s shell manufacturing, the infill value is 100%, in the FEA analysis, it was considered that the specimen was working like a single block and has the same properties in the entire volume. If it is considered that the specimen’s shell has other properties than those of the central area, then, through tests, these must be established. In this case, it will be considered that the specimen is composed of two or more zones (entities), depending on printing parameters, and the simulation process will be performed in the same way as for the multi-material specimens.

Two simulation methods were applied for this type of specimen. The first method used the rules of mixture (ROM) [20] to calculate the material mechanical properties of the bending model. In this case, the equation for the calculation of the property values of the material obtained by combining two types of filament is Equation (1):P_c_ = P_m_·V_m_ + P_p_·V_p_(1)
where P_c,m,p_ represents the property value, V_m,p_ represents the volume fraction, and c, m, p represent the composite, matrix, and phase. The Equation (1) was applied to determine the specimen material properties: density, yield stress, and modulus of elasticity. In the case of specimens obtained by combining filaments, the proportion of material for the outer layer was 31%, and that of the inner layer was 69%. This proportion is valid for both circular section and elliptical section specimens (see Figure 8).

For the second method (called SIM [19]) applied for bending simulation of the combined specimens, it was considered that the specimen is composed of two semi-specimens (one that constitutes the exterior and another that represents the interior). For each of the two semi-specimens, the key material properties were assigned, corresponding to each filament from which they were made. By assembling the two semi-specimens, the model of the specimen is obtained, which is considered to work as a block. For the two described methods, the bending simulation process was applied, considering that the bending loading (force) of the model was the one registered in the bending tests. The results obtained by using the two methods are presented in Figure 9 and Figure 10, and Table 3. The analysis of the table shows that, for this type of test, the results of the simulation are close to those of the bending tests.

The analysis of Table 3 shows that, for the strength characteristic, by applying ROM method, three out of four results have errors that are less than 5%, comparatively with the tests’ values. For the total deformation characteristic, the analysis of Table 3 shows a similar behavior with strength evolution. The error value is greater, but only for one specimen (GP_E18 by ROM method). The tests show that specimens allow greater deformation in comparison with those calculated by the simulation process. Moreover, a few values obtained by simulation are smaller than those recorded in the tests.

## 4. Conclusions

The studies presented in the paper aimed to show if the simulation process of 3D-printed PLA structures, when these are bending stressed, allows viable results to be obtained. Viable results mean that the values of the mechanical properties or characteristics analyzed are close to the values of the same properties (characteristics) established by bending tests. Another direction of the study was to show that the theories valid in the strength of materials and applicable to metallic or non-metallic materials have similar effects for 3D-printed PLA structures. This was the reason why the study was done for both circular or elliptical cross-sections and solid or ring-shaped sections (bar- or tube-type section). The paper shows that, for the additive manufactured specimens from the studied filaments, or by combining them, the results of the simulation process are close to those of the bending tests. For the bending stress, in the case of bar-type specimens, the deviations of the simulation results compared to those of the tests are between 7.2 and 5.7%. In the case of bending deflections (displacements), these deviations are between 17.3 and 8.5%. For tube-type specimens, the differences between test and simulation results are between 1.8 and 7.3% for bending strength, and, for the displacements, the simulation values are smaller than the test values. In the case of 3D-printed specimens obtained by combining two filaments, the deviations’ average is around 5% (depending on the simulation method) for bending strength, and around 10% for the displacements.

Referring to efficiency of the material used, established by the ratio between the ultimate bending strength and cross-section value, the best solution for the structure configuration, obtained for additive manufactured PLA, is for the tube-type specimen with an elliptical section.

The paper also shows that, for PLA 3D-printed structures, the geometry of the cross-section has an important role in the judicious use of the material. The analysis of Table 2 and Table 3 shows that the most efficient configuration for the specimen is the tube-type with an elliptical cross-section (P_E18_T code). Moreover, for all specimens, those with an elliptical section have a better behavior at bending, comparatively with those that have a circular section.

The final conclusion is that the simulation process can be applied with good results on the 3D-printed PLA structures when bending stressed. The results of the simulation are in line with those of the bending tests, but it should be emphasized that their accuracy depends on the model used in the simulation process, more precisely, how this reproduces the 3D-printed structure. The accuracy of the results also depends on the simulation software’s capabilities. In addition, the simulation cannot be applied with acceptable results if the key material properties of the additive manufactured structures and the parameters used during printing are not known. On the other hand, the paper provides the premises in order for the simulation process to be applied with good results in the optimization of the 3D-printed PLA structures. Moreover, knowledge of the strength of the material can be applied with good results to study the behavior at mechanical stresses of the additive manufactured structures. For the specimens obtained by combining two filaments, it was observed that, when the PLA–glass filament is arranged on the outside, the structure’s deformation capacity improves.

## Figures and Tables

**Figure 1 molecules-26-03325-f001:**
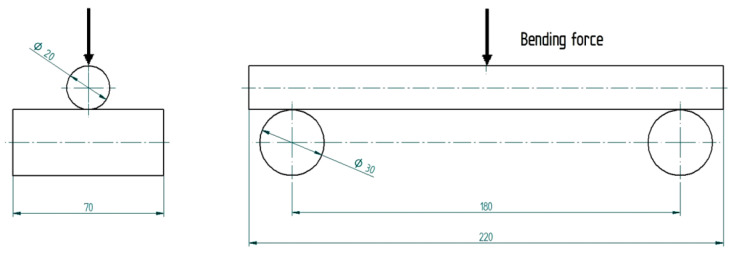
Bending scheme (specimen with circular cross section).

**Figure 2 molecules-26-03325-f002:**
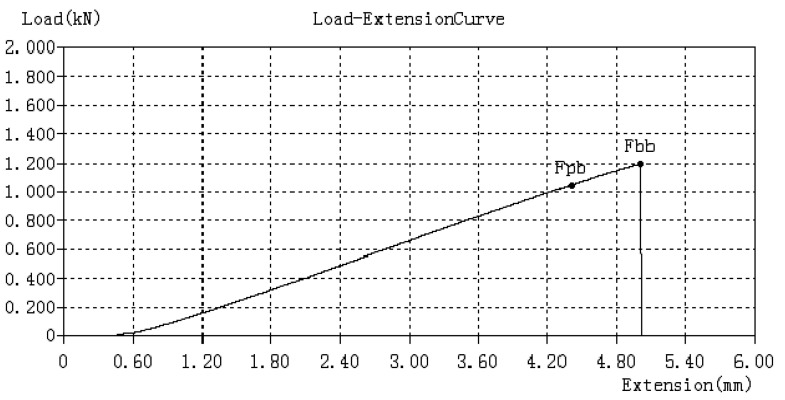
Load–extension curve obtained during bending test (P_24 specimen).

**Figure 3 molecules-26-03325-f003:**
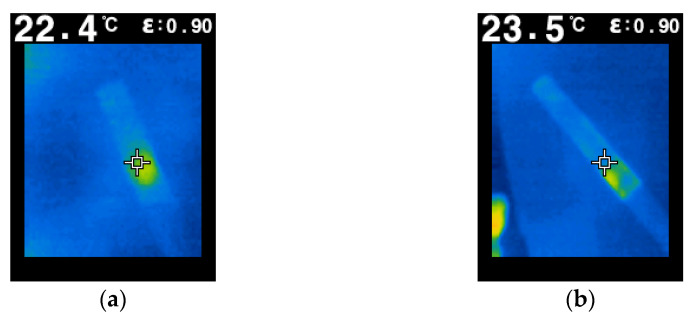
Temperature evolution during bending tests: (**a**) P_24; (**b**) G_24.

**Figure 4 molecules-26-03325-f004:**
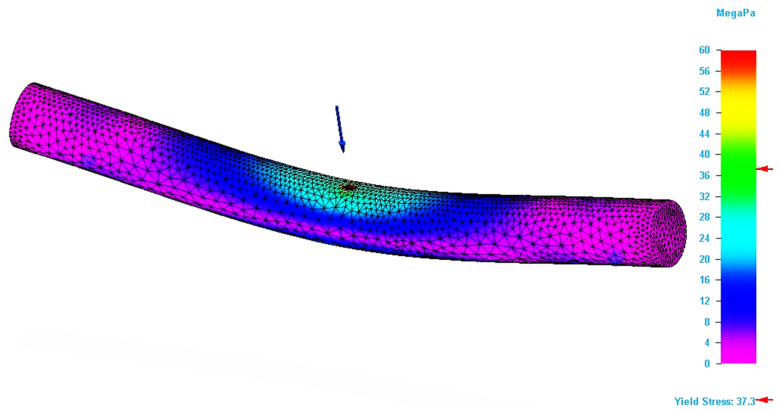
Result of the simulation process for P_20 specimen (Von Mises stress).

**Figure 5 molecules-26-03325-f005:**
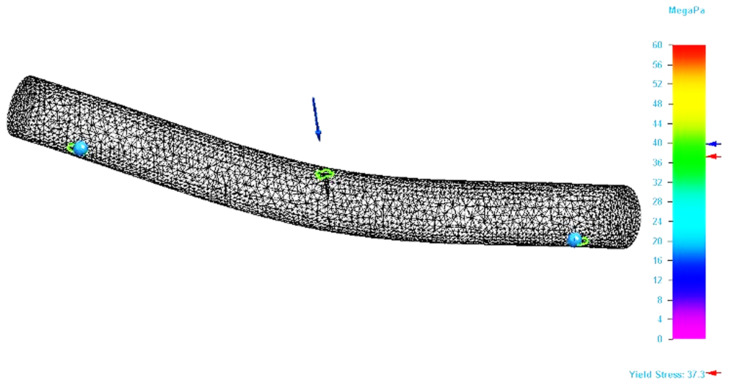
Specimen zone where stress value is equal with ultimate bending strength (P_20 specimen).

**Figure 6 molecules-26-03325-f006:**
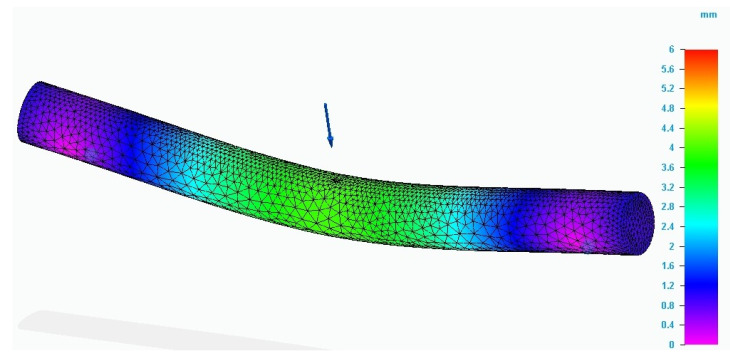
Result of the simulation process for P_20 specimen (total deformation values).

**Figure 7 molecules-26-03325-f007:**
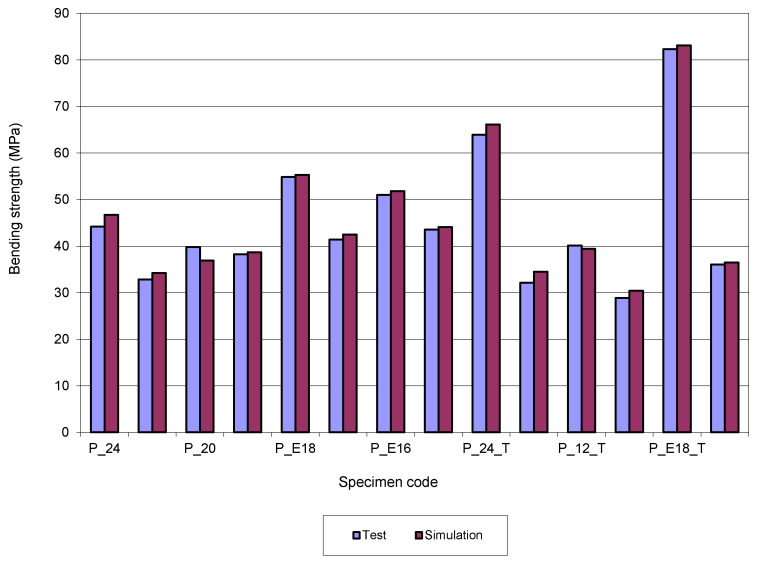
Comparison between bending strength values for 3D-printed specimens.

**Figure 8 molecules-26-03325-f008:**
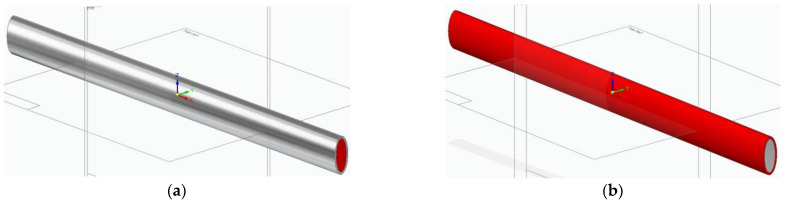
Combined 3D-printed specimens—elliptical cross section: (**a**) PLA + PLA–glass; (**b**) PLA–glass + PLA).

**Figure 9 molecules-26-03325-f009:**
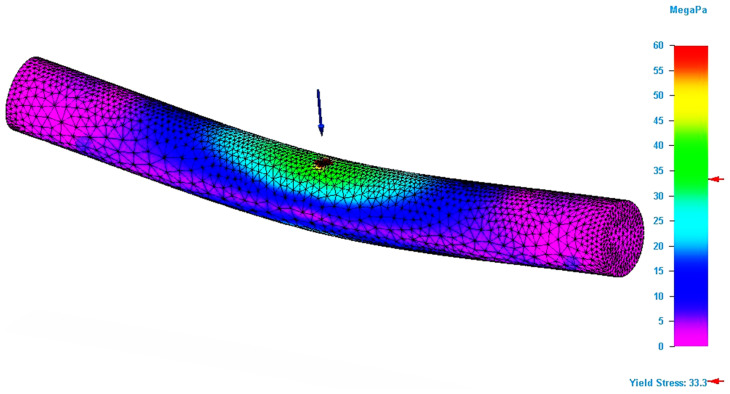
Result of the simulation process (ROM method) for PG_24 specimen (Von Mises stress).

**Figure 10 molecules-26-03325-f010:**
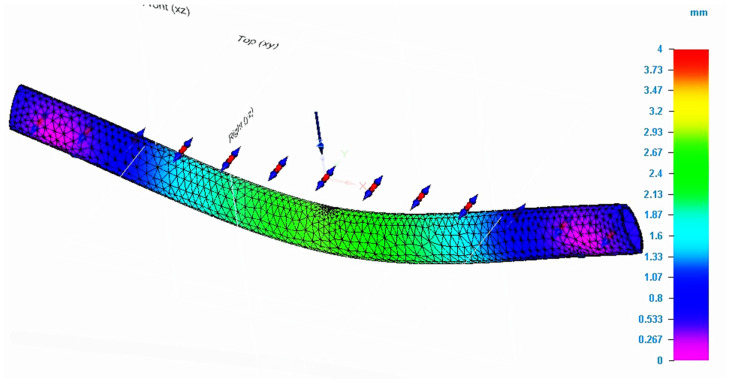
Result of the simulation process (SIM method) for PG_E18 specimen (total deformation values).

**Table 1 molecules-26-03325-t001:** Specimen characteristics and codification (coding).

Filament Type	Cross Section	Dimensions (mm)	Specimen Type	Specimen Code
PLA	Circular	24	Bar	P_24
PLA–Glass	Circular	24	Bar	G_24
PLA	Circular	20	Bar	P_20
PLA–Glass	Circular	20	Bar	G_20
PLA	Ellipse	18 × 8	Bar	P_E18
PLA–Glass	Ellipse	18 × 8	Bar	G_E18
PLA	Ellipse	16 × 6	Bar	P_E16
PLA–Glass	Ellipse	16 × 6	Bar	G_E16
PLA	Circular	24 × 20	Tube	P_24_T
PLA–Glass	Circular	24 × 20	Tube	G_24_T
PLA	Circular	12 × 10	Tube	P_12_T
PLA–Glass	Circular	12 × 10	Tube	G_12_T
PLA	Ellipse	18 × 16	Tube	P_E18_T
PLA-Glass	Ellipse	18 × 16	Tube	G_E18_T
PLA + PLA–Glass	Circular	24	Bar	PG_24
PLA–Glass + PLA	Circular	24	Bar	GP_24
PLA + PLA–Glass	Ellipse	18 × 8	Bar	PG_E18
PLA–Glass + PLA	Ellipse	18 × 8	Bar	GP_E18

**Table 2 molecules-26-03325-t002:** Comparison between results of the tests and simulation for 3D-printed specimens, bending stressed.

Specimen Code	Test Results		Simulation Results	
	Strength (MPa)	Deformation (mm)	Strength (MPa)	Deformation (mm)
P_24	44.18	4.92	46.70	4.98
G_24	32.84	3.5	34.20	3.70
P_20	39.78	5.54	36.90	4.77
G_20	38.26	4.95	38.70	4.09
P_E18	54.85	5.81	55.30	6.10
G_E18	41.40	4.11	42.50	4.46
P_E16	51.02	6.05	51.80	5.83
G_E16	43.57	4.78	44.10	4.44
P_24_T	63.94	7.5	66.10	8.51
G_24_T	32.16	4.08	34.50	3.94
P_12_T	40.14	7.54	39.40	5.85
G_12_T	28.87	7.75	30.40	3.76
P_E18_T	82.33	6.37	83.10	8.17
G_E18_T	36.07	3.37	36.50	3.22

**Table 3 molecules-26-03325-t003:** Comparison between results of the tests and simulation for 3D-printed specimens, bending stressed (two filaments used).

Specimen Code	Test Results		Simulation Results			
			ROM		SIM	
	Strength (MPa)	Deformation (mm)	Strength (MPa)	Deformation (mm)	Strength (MPa)	Deformation (mm)
PG_24	49.82	4.54	52.05	4.70	52.30	4.37
GP_24	46.90	4.22	51.80	4.43	49.00	4.14
PG_E18	21.23	3.08	22.20	2.36	22.90	2.78
GP_E18	34.32	7.65	35.90	5.35	35.10	7.94

## Data Availability

Not applicable.

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
