# Peer review of "Comparison between the Test and Simulation Results for PLA Structures 3D Printed, Bending Stressed"

_molecules, 2021, doi:10.3390/molecules26113325_

Round 1

Reviewer 1 Report

Manuscript entitled “Comparison between the test and simulation results for PLA structures

3D printed, bending stressed” has been carefully read and analyzed in context of publication in the MDPI journal MOLECULES.

The manuscript is interesting and presents a fair level of expertise in the area of polymer sciences. I suggest accepting it for publication after some improvement.

Below you can find list of my suggestions.

  1. First of all, I would like to complain that the English of this manuscript require thorough revision. In current state the article is difficult to read.
  2. The Introduction part is quite long and cover broad range of the investigation background. However It is lacking one very important part i.e. The explanation of the objectives of the woks and indication of its significance and the novelty.
  3. Fourth paragraph of the Introduction contain the statement: “... polylactic acid can replace the ABS because it is totally eco friendly ...”. The sentence is uncritical repetition of the manufacturer slogan, the scientists should be more critical in their opinions and know that ABS is not biodegradable neither obtained from renewable sources.
  4. Another thing that I would like to raise is the name of the polymeric material author use “polylactic acid” according to UIPAC its name should be written poly(lactic acid)
  5. Using abbreviations is common and reasonable practice, however introduction of an abbreviation require to define it and the first place where it is used. For example:

-  the abbreviation for finite element analysis (FEA) is defined in the keywords, but it should be also indicated when it first occurs in the text instead of doing this on the 6 page after fourth use.

- The abbreviation ABS is used without any explanation. I understand that this is common abbreviation of the material name but this is scientific paper and it should fulfill some standards.

Author Response

Dear reviewer,

First of all, thank you for reviewing the manuscript.

I have taken note of the observations made on the manuscript with title “Comparison between the test and simulation results for PLA structures 3D printed, bending stressed“. We modified it in accordance with the recommendations made. We explained the abbreviations that appear in text, including for FEA. In the case of FEA we explained the abbreviation at its first meeting in the text.

Sincerely yours,

Authors

Reviewer 2 Report

In my opinion, the work does not seems aligned with the aim of the journal. Furthermore, it does not present elements of innovation, neither in relation to materials adopted, nor in relation to modeling. The comparison with experimental tests, announced by the title, should be improved. Statistical analysis should be added.

Author Response

Dear reviewer,

I write to you regarding the manuscript entitled “Comparison between the test and simulation results for PLA structures 3D printed, bending stressed”.

I read the reviewers' recommendations, that is why I implemented them in the manuscript. 

Sincerely yours,

Authors

Reviewer 3 Report

The article "Comparison between the test and simulation results for PLA structures 3D printed, bending stressed"  presents a comparison between the results of the bending tests and simulation of PLA and PLA-Glass with different geometries and forms. The article is well written, but some changes need to be performed before publishing it.

The introduction need more references. For instance, in 2nd page paragraph 1 line 5 or in teh paragraph 2. Moreover, in the 3rd paragraph the words FEA and SBF are not explained, as well as HDPE in page 3.

The experimental part is well explained, but in the case of the printing process the parameters must be explian in a paragraph or including a table.  On the other hand, the figure 1 is not well explained and it is not clear enough and the Figure 2 is an old format graph which must be changed.

In the results the Figure 3 needs more explanation in the text. On the other hand, the simulation properties are explained in the 6 page of the results, and these parameters must be mention in the experimental part. On the 4th paragraph of page 6 references must be written. Figures 5 and 6 can be put together. Moreover, take out the table included in Figure 5, it is not neccesary and it is hard to read it. Also, it is difficult to understand why in test and in simulation the same strength is not used in both analysis.  Moreover, with this mentioned table the Figures 7 and 8 do not give any more information, so they are not neccesary.  In the page 10, in the 2nd paragraph  the methods required references.  The same happens in the last paragraph of page 10 abput the SIM method. The figures 12 and 13 are not important because the data is presented in Table 3. They should be included in the supporting information.

The conclusions and references are well written.

Author Response

Dear reviewer,

Thank you for the recommendations made to improve the manuscript with title “Comparison between the test and simulation results for PLA structures 3D printed, bending stressed”.

We have implemented your requirements.

Thus, for figures 1 and 3 we explained their significance. We also explained why figures 2, 5 and 7 appear in the manuscript.

We removed figures 8, 12 and 13. In addition, we moved a paragraph from Results and discussions to Experimental setup.

Sincerely yours

Authors

Round 2

Reviewer 2 Report

The work remains not aligned with the aim of the journal. It does not present elements of innovation, neither in relation to materials adopted, nor in relation to modelling.

The comparison with experimental tests has been improved and included in the manuscript. 

On page 3, lines 101-103, the authors declare "Researchers are trying to find solutions to problems that appear during the printing and use of 3D printed poly(lactic acid) structures. These concerns are present in numerous scientific articles." Some references should be added, also focusing on recent advancements in Additive Manufacturing (AM) technologies and on the possibility to combine different technological strategies and advanced materials (see for example De Santis R. et al., Materials 202114(1), pp. 1–15, 181) or the possibility to modify/decorate structure surfaces - adopting different strategies - for improving functional and structural features (see for example Russo L. et al Carbohydrate Research, 2015, 405, pp. 39–46; Russo L. et al RSC Advances, 2013, 3(18), pp. 6286–6289).

On page 6, lines 304-306, the authors declare "In other words, PLA and PLA-Glass 3D printed parts behave similarly to alloys, which mean that certain concepts valid for alloys can be used with good results for the 3D printed parts." The concept needs to be further clarified with supporting references.

The authors should also clarify how the proposed model could support the design of more complex 3D structures, for example by focusing on complex 3D lattice structures. To this aim, the entire manuscript, including the conclusion section, should be improved focusing on the possibility to adopt the obtained results for prediction of other parameters for further modelling analyses (see for example Maietta S. et al, Materials 2018, 11(2), 312).